# Ageing and Loneliness in Times of Pandemic: A Scoping Review

**DOI:** 10.3390/ijerph20075337

**Published:** 2023-03-30

**Authors:** Raúl Ruiz-Callado, Diana Jareño-Ruiz, María Elena Fabregat-Cabrera, María Manuela Penalva-Lorca

**Affiliations:** 1Department of Sociology I, Faculty of Economics and Business, Alicante University, San Vicente del Raspeig, 03690 Alicante, Spain; raulruiz@ua.es (R.R.-C.);; 2GIS-UA, Faculty of Economics and Business, Alicante University, San Vicente del Raspeig, 03690 Alicante, Spain

**Keywords:** ageing process, loneliness, COVID-19, social relationships, social distance, person-centred care, family solidarity, active ageing, scoping review

## Abstract

Society is immersed in a process of demographic transformation of great relevance: the ageing of the population. During the ageing stage, relevant changes occur, such as age-related losses, lack of formal and informal support or changes in social roles, which can cause situations of isolation or loneliness in older people. After the Spanish government decreed a state of alarm throughout the national territory in response to the arrival of SARS-CoV2, the confinement of the entire population was established, and only essential services and authorized persons could continue to carry out their daily activities and professional tasks. In addition, since the beginning of the pandemic, older people were considered high-risk people, a circumstance that increased their isolation situation. Objective: Understand, organize and systematically analyse the scientific evidence generated in relation to the impact that the COVID−19 pandemic has had on the processes and feelings of isolation and loneliness of the elderly population, from the beginning of the health crisis until the date of search. Materials and methods: a scoping review was conducted using the methodology of Arksey and O’Malley, which included a review of the studies available in the online databases Proquest, Scopus and WOS. From the search, information related to the isolation and collective of elderly people during the pandemic was extracted. Results: a total of 32 articles were included, from which three fundamental areas of analysis emerged and three issues emerged: older people and institutionalisation during the pandemic, ageism and hospitalisation of the elderly during the pandemic, and loneliness and isolation of older people throughout the pandemic. Discussion: the lack of material resources and infrastructures to be able to face the problem of isolation in institutions was evident. The importance of acquiring, on the part of the elderly, competences, knowledge and skills in new technologies in order to continue with contact with their peer group and family was also observed. Conclusions: this study identifies areas already understood, as well as knowledge gaps, that allow for determining opportunities for future research and thus the ability to improve in situations similar to the one that occurred.

## 1. Introduction

Society is immersed in a process of demographic transformation of great relevance: the ageing of the population.

In the 1960s, different theories emphasised the need to reduce the participation of older people as their age increased until their definitive retirement [1]. On the other hand, some authors framed within their theory of activity that people within their ageing process should acquire new habits to achieve well-being and continue with activities focused on this new stage [2]. Within the idea of continuity of the elderly, it is confirmed that the knowledge acquired throughout the life process is crucial to carry out personal strategies to face in a positive way the changes that have occurred in this vital stage [3].

During the aging process, older people must be able to adapt to the loss of roles and identity. In order to face these changes, a process must be carried out in which integration, social participation and the continuity of a social network, formal or informal, is the main strategy to avoid situations of isolation, loneliness or vulnerability.

Some authors define emotional loneliness and social loneliness, where they emphasise the individual understanding of people and how cognitive processes such as self-esteem or social skills influence its management. The elderly express loneliness through two dimensions, the objective dimension being the one in which their concerns focus on the lack of protection against possible supervening health situations, while on the other hand there is the material dimension, this being more subjective and related to the feelings of the person, the absence of relationships and lack of support [4].

It was in March 2020 when the government of Spain decreed the state of alarm. One of the measures adopted was the confinement of the entire population to their homes. Since the beginning of the pandemic, older people were considered high-risk people. The confinement produced a complex situation for the elderly since many of them were, before the pandemic, integrated into activities with peer groups or developing rehabilitation activities, which were cancelled to avoid direct contact with other people; even formal and informal care ceased to be received.

The absence of both rehabilitation services and essential services aimed at the elderly meant the need for continuity of care by formal and informal care workers. Care demands are usually related to the late onset of deterioration of people’s health. That is why, after the closure of rehabilitation services and the absence of direct contact with other people, there was an increase in help within the family. María Ángeles Duran emphasises the importance of fostering intergenerational relationships because there is a large number of single-family households [5]. That is why family has such an important role as a source of material and emotional support during the pandemic.

In the majority of the population, it is only those people with a high level of dependency and, sometimes, with the absence of close sources of formal or informal help, who require institutionalisation in order to cover their basic needs.

By compiling the papers selected after searching different databases, this study aims to find out the impact that the pandemic had on the isolation and loneliness of the elderly. 

## 2. Materials and Methods 

To achieve the objectives of the research, the methodology that has been applied is scoping review [6], with the aim of synthesising the knowledge around the specific research question via the selection and systematic synthesis of literature, in order to obtain greater conceptual clarity on a topic or specific field of research.

We followed the stages proposed by Arksey and O’Malley, which include:I.Identification of the research question.II.Identification of relevant studies.III.Selection of appropriate studies.IV.Data logging.

Step 1: 

The research question that guided the scoping review was: What were the consequences of the pandemic on older people’s experiences, routines, loneliness and isolation during confinement?

Step 2: 

The identification of the articles was carried out through a systematic search process in the databases Proquest, Scopus and WOS. The identification of the keywords according to our research question, allowed us to specify the following search equation.

(isolation OR Loneliness OR Solitude) AND (elder* OR “old age” OR “old people”) AND (social and (life OR action OR participation or contact or interaction) OR “active aging” OR “active ageing”) AND (COVID OR SARS-CoV OR SARS-CoV-2 OR coronavirus)

The keywords were limited to the sociological approach to the ageing process developed in this contribution. From the theory suggested by Atchley in 1971 [3], where he asserts that the knowledge acquired throughout people’s life processes is crucial to being able to carry out personal strategies to cope positively with the changes that occur during ageing, the need to add keywords such as participation, contact, interaction was observed, as these are related to the subject under study.

The search was carried out on 2 February 2022 against the three databases mentioned above. There were three inclusion criteria applied in the search, as can be seen in Table 1.

The first of them refers to the language in which the search was carried out; the second criterion was related to the time in which they had been published (the publications of the five years prior to the search were selected); and, finally, those publications in scientific journals, doctoral theses, reports and proceedings, books, reports, and state publications were included.

Step 3: 

The results obtained were imported to the bibliographic reference manager Refworks, a space from which duplicates were identified and eliminated. The selection of the articles was based on the reading of the titles, abstracts and texts in which the keywords appeared. Figure 1 below shows the analysis carried out for the selection of articles.

Step 4: 

To standardise the process of extracting relevant data, a summary was made that included the following categories of information: author(s) and year, subject, sample, methodology and results or main findings. 

Step 5: 

To identify the topics addressed in the studies, the extracted information was collected and categories related to our area of interest were assigned. The following Table 2 shows the main methodological strategies used in the studies analysed.

According to the results obtained in the synthesis process, three main issues emerge during the pandemic:Older people and institutionalisation during the pandemic.Ageism and hospitalisation of the elderly during the pandemic.Loneliness and isolation of the elderly throughout the pandemic.

## 3. Results

Then, through the following diagram (Figure 2), the most relevant information was collected, which will be analysed in more detail in the following subsections.

### 3.1. Elderly People and Institutionalisation during the Pandemic

Some of the studies analysed reveal the impact of the pandemic, especially on institutionalised older people and those living in single-family households.

In Spain, the National Institute of Statistics carried out the collection of information on the structure of households in the country to learn about the permanence of older people during confinement in their homes. According to data collected in 2019 in Spain, one in four households were single-person; this means that almost 4.8 million people spent confinement alone. In the population and housing censuses it was observed that the population living in collective establishments increased by 90.3% [7]. In the data provided by the population and housing census of 2011, the predominance of women in residences accounted for 65.2% of people living in them, that is, a higher percentage compared to the number of institutionalised men; 82.6% of women and men were 65 years or older [8].

According to the research reviewed, the initial protocols that were carried out in the long-stay centres were based on a rapid diagnosis that involved isolation and individual intervention by professionals; however, the this necessary isolation and the avoidance of contact with people who needed quarantine could not be fulfilled due to the lack of both material and human resources and infrastructure. All this caused long-stay centres to become a source of contagion [9].

Long-stay centres are also considered day centres or rehabilitators. The closure of these services with the arrival of the pandemic meant that people who came to these services had to stay at home. The formal and informal support that the elderly received were altered during the state of alarm, having to adapt according to the needs of each person [10].

According to the National Institute of Statistics, 71.62% of the persons aged 65 and over are women, while 28.38% are men. According to the studies reviewed, isolation during the state of alarm meant a decrease in people’s social and family interaction, causing a greater feeling of loneliness and abandonment [11]. In Spain, the contribution of families to the welfare of their members remains of great importance, a main source of services and care. The arrival of the pandemic meant an alteration in the care system for the elderly: there was deterioration in terms of the quality of care for people who lived alone and needed care, in addition to those elderly people who also required attention.

Table 3 shows the selected articles that helped in the development of this first subsection.

### 3.2. Ageism and Hospitalization of the Elderly during the Pandemic

The World Health Organization defines health equity as “the absence of unfair and avoidable or remediable differences in health or between socially, economically, demographically or geographically defined population groups” [17]. In different regions of Spain, some health areas and departments established action protocols that did not comply with international ethical standards [18].

The term ageism refers to discrimination against older people, either by stigmatisation or discriminatory practices by society and public or private institutions. Health professionals had to prioritise when intervening, with chronological age being one of the main factors to be taken into account [19].

The lack of initial resources in the face of the health emergency led to the establishment of criteria such as not caring for those dependent older people. The possibility of life expectancy of young people prevailed over the good functional capacity of the elderly, who were being barred from receiving attention only because of their age. Those people who died because of their condition of being “elderly” did so alone, and therefore the families denounced the deaths of people affected by the virus who did not receive medical attention to be able to cope with the disease.

In April 2020, Age Platform Europe published a report on the situation of older people, which stated that older people had the same right to be protected during the pandemic, so age should not have been a criterion for medical triage [20].

Table 4 shows the selected articles which helped in the analysis and development of this subsection.

### 3.3. Loneliness and Isolation of the Elderly throughout the Pandemic

During the state of health emergency, the elderly were the main victims, since they were considered the most vulnerable group to the spread of the virus. Social distancing meant that those people who lived in their homes had no contact with their formal and informal social network; even their rehabilitation treatments were paralysed, causing a decrease in their quality of life [12].

Studies that analysed the experiences of older people who age in their place of residence show the perception of this group in the first person. These studies sought to know the situation of elderly people affected by COVID-19 [24,25].

Many of the people interviewed in the study showed a high degree of anxiety and stress due to ignorance of the situation. In the same way, many of the relatives who cared for elderly people at home perceived discrimination when receiving treatments, which caused situations of loneliness, since health workers stopped going to homes being consultations via telephone.

The elderly people interviewed emphasised the importance of being able to participate in medical decisions once they were hospitalised due to their worsening health. They considered the information received at all times about the situation to be very helpful and thus be able to feel protected from a system that had initially caused them a situation of fragility and isolation.

The isolation caused by confinement occurred, as we have previously pointed out, both at the institutional level and in people’s homes. The capacity for resilience and empathy was important to be able to face the situation they were in, as will be detailed below. The studies show how hospitalised elderly people responded to the lack of support, that is, in the absence of family members or careworkers altruistically providing help to older patients who were in worse conditions. This help was provided by giving technological communication means to those who did not have them, so that they could communicate with their families [13].

Access to information and communication technologies was not available to all older people, either due to ignorance of their use or the lack of the necessary tools to be able to relate to their environment. According to the National Institute of Statistics, in 2021, 31.8% of people over 75 years of age used ICTs in relation to the first period of the pandemic, which was 27.9% of the population over 75 [14]. These data show the scarcity of means available to them to continue contact with those closest to them.

Table 5 shows the selected articles which helped in the analysis and development of this subsection.

## 4. Discussion

This research aimed to collect the scientific evidence produced from the beginning of the pandemic to the date of research in relation to the impact it has had on isolation, loneliness and experiences of ageism of older people.

From the results obtained in this study, it can be deduced that the situation of the ageing population during and after the pandemic, taking into account the confinement, deteriorated by the limitations imposed, creating an even greater sense of vulnerability in this group. Likewise, the sudden arrival of the pandemic made apparent the scarcity of both human and material resources. Another fact to keep in mind is that the elderly were considered high-risk citizens as well as the most fragile fighting the virus; all this caused them to be excluded, from the outset, from medical assistance if they suffered complications caused by the infections. Similarly, isolation meant an interruption of interaction with their families and peer groups, thus causing situations of insecurity. This shows that society was not prepared for an event of such magnitude and essential services must be constantly updated in order to be prepared for the possible arrival of similar situations.

It can be seen that many of the studies analysed were carried out in a short period of time so they present a very superficial vision of the problem in which we find ourselves. They will be the investigations closest to the date of our search, which give us more complete information and knowledge of the experiences of the elderly group during the pandemic. It should be noted that some supervised systematic reviews deal with quantitative studies having as their main objective the study of the prevalence of loneliness where the importance of conducting longitudinal studies is highlighted in order to observe the long-term impact of the pandemic [31,32]. In addition, based on the results obtained, it is considered that furtherer research should be conducted to improve the situation of this group regarding issues such as access to new technologies and its operation, the provision of care training for family members and care workers, and the observation of the various deficiencies that have been suffered during the pandemic, and thus begin to establish action protocols in future similar situations.

## 5. Conclusions

The purpose of this scoping review, as previously indicated, has been to carry out an exploratory review of the literature, to determine the current state of the research in relation to the impact that the pandemic has had on the experiences of isolation and loneliness in the elderly population. The conclusions obtained have made it possible to identify knowledge gaps and new research opportunities in relation to the topic, allowing a deeper understanding of the phenomenon and making lessons emerge so that better management of similar situations can be possible in the future [26]. Both older people who are living in public or private institutions, as well as those who do not have any formal or informal social network with functional or cognitive limitations, were faced with a situation of helplessness and uncertainty. It is important to remember that the change in the care model caused alterations within the health field: all visits to residential and hospital centres were restricted, thus causing a greater sense of isolation and a great change in end-of-life care where family and relatives were absent from direct contact with people throughout the disease process, as seen in the study by Manzor and Hamid [15]. Similarly, people who lived in single-family homes, although they did not require care for the most part, were excluded from daily physical contact, that is, they were isolated, causing them emotional problems.

It was at this stage where the use of new technologies to maintain contact with relatives increased. Some authors emphasise the importance of using ICTs to maintain social and family contact. For all these reasons, it is important to encourage the use of new technologies (video calls and social networks) as they can avoid situations of loneliness and isolation due to lack of real contact [27].

Also, it is worth highlighting some studies that reflect the importance of the participation of older people in decision-making [33]. That shows the importance of people-centred care, since they can be excluded if they are not participants in the activities of society despite wanting to carry them out. Therefore, the views of older people are essential in order to understand their experiences and thus to be able to promote further improvements in the field of care, such as developing new disease prevention and health promotion strategies [21]. The participation of older people and their care workers in the design of an adequate environment for coexistence, continuous training in long-term care, as well as the administration of new technologies to avoid isolation, will help increase the quality of life in long-stay and/or short-stay centres [16,22,28].

According to the studies analysed, public administrations must pay more attention to vulnerable groups, in addition to strengthening health systems and improving working conditions in order to carry out family reconciliation in future situations similar to that of the pandemic [29]. The authors of the reviewed articles propose, in turn, ideas for improvement for the management and coping with possible similar scenarios.

In summary, the main points to highlight after carrying out this review are the following:Bet on new technologies, that is, publicise and make accessible ICTs to the whole group as this will favour the continuity of a “new normal”. In short, eliminate the existing digital divide in the elderly [23,34].Support and take into account the opinions of older people in order to be able to adapt the work to the demands actually expressed [30].Develop an adequate environment in long-stay centres, that is, centres that have adequate infrastructures to be able to create safe places before possible isolations, thus favouring the health of the elderly.Train formal or informal workers and care workers for the prevention and control of future similar situations and adapt protection resources to them.Recognize the right to health and equitable care integrity for all groups, since restrictions based on the chronological age of the person mean that the real capacities of the person are not valued.Work on care information systems that provide real-time testimony from people who are alone and may suffer falls or emergency situations.

## Figures and Tables

**Figure 1 ijerph-20-05337-f001:**
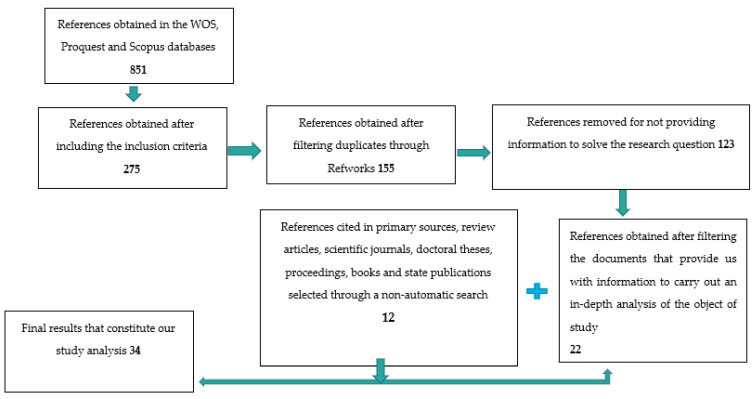
Summary of the workflow under review and number of items retrieved, deleted and selected. Source: own authors.

**Figure 2 ijerph-20-05337-f002:**
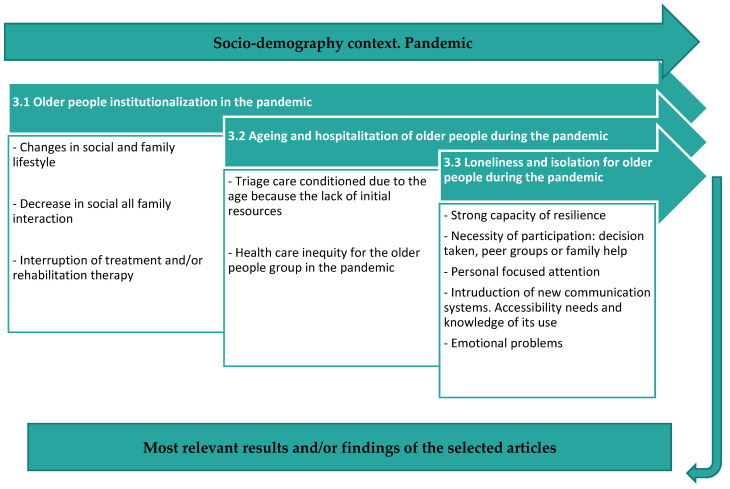
Diagram of the situation during the pandemic of the elderly according to the selected articles. Source: own authors.

**Table 1 ijerph-20-05337-t001:** Inclusion criteria. Source: own authors.

Inclusion Criteria
Languages: English and Spanish
The Works are Published within the Last 5 years
Type of Source: Scientific Journals, Doctoral Theses, Reports and Proceedings, Books, Reports and State Publications.

**Table 2 ijerph-20-05337-t002:** Methodological strategies used in the selected articles. Source: own authors.

Methodological Strategies
Systematic Revision
Situational Analysis
Case Analysis
Questionnaires
Surveys
Self-ethnography
Statistic Reports
Systematisation and Analysis of Statistical Sources
Social Network Analysis
Interviews
Surveys (telephone and face-to-face)

**Table 3 ijerph-20-05337-t003:** Articles selected according to area and topic of interest. Source: own authors.

Older People and Institutionalisation during the Pandemic				
Authors and Year	Subject	Sample	Methodological Strategy	Results or Main Findings
National Institute of Statistics (INE). 2011 [7]	Population analysis.	Survey in all collective establishments in the national territory	Survey	Relative tables and population figures, statistical tables on the characteristics of residents.
Instituto Nacional de Estadística (INE) 2020 [8]	Population analysis	Continuous household survey (ECH).	Survey	Relative tables and population figures, statistical tables on the characteristics of residents.
Pastor Seller. E. 2021 [11]	Family, social change, COVID-19, vulnerable families, social protection, social services and family policies	Changes in family structure and behaviours.	Systematisation and analysis of statistical sources.	Social policies for the protection and support of families are scarce and insufficient to offer guarantees of care within the family.
Izatari, M. Risco, E. Cesari, M. Buurman, BM. Kuluski, K. Davey, V. Bennett, L. Varela, J and Pruu Bettger, J. 2020 [12]	Health, institutionalisation, COVID-19 and the elderly.	Elderly people during confinement.	Case analyses	Address the long-term care of institutionalised people in order to create optimal standards for the quality of life of these people.
Johanna Gustavssoon and Linda Beckam. 2020 [13]	Seniors, mental health, COVID-19, health perception	People over 70 years old.	Survey	Older people comply with the measures imposed. Isolation can lead to worsening mental health and long-term effects.
Auyeung, Tung. Chan, Felix. Chan, Ty. Lee, Jenny. Luk, James. Mok, Winnie. Shum, Chun Keung. Wong, Cw. 2020 [14]	COVID-19, frail elderly, infection control.	Older people	Experiences in caring for the elderly during the pandemic.	Deterioration of care. Reflection on the opportunity to develop a new service model.
Mohammad S. Razai. Pippa Oakeshott. Hadyn Kankam. Sandro Galea. Helen Stokes-Lampard. 2020 [15]	Effects of isolation, COVID-19, continuity of care, social conditions, psychological well-being.	Older people	Surveys	Assessing the psychological effects of social isolation during COVID-19.
Judith R.L.M Wolf y Irene E. Jonker. 2020 [16]	Empowerment, person-centred care, recovery, social exclusion, quality of life.	Older people	Person-centred intervention program.	Review of the definition of quality. Help develop factors and resources for care.

**Table 4 ijerph-20-05337-t004:** Articles selected according to area and topic of interest. Source: own authors.

Ageism and Hospitalization of the Elderly During the Pandemic				
Authors and Year	Subject	Sample	Methodological Strategy	Results or Main Findings
Matteo Cesari and Marco Proietti. 2020 [19]	Health, seniors, COVID-19.	Older people in the health system.	Situational Analysis.	Absence of criteria to avoid ageism. Medical triage is motivated by age without taking into account the functional capacity of people.
Johanna Gustavssoon and Linda Beckam. 2020 [13]	Seniors, mental health, COVID-19, health perception	People over 70 years old.	Surveys.	Older people comply with the measures imposed. Isolation can lead to worsening mental health and long-term effects.
Guiomar Merodio. Mimar Ramis-Sola. Diana Valero. Adrian Ausbert. 2020 [21]	Age discrimination. Equitable healthcare, COVID-19, elderly people, human rights.	Elderly people hospitalised and recovered from COVID-19. Family members and caregivers of the elderly and health professionals.	Qualitative interviews	Important as well as transformative aspects related to family relationships, solidarity actions and humanized care are observed.
Sarah Fraser, Martine Lagacé, Bienvenu Bongué, Ndatté Ndeye, Jessica Guyot, Lauren Bechard, Lidia Garcia, Vanessa Tales, CCNA Social Unclusion and Stigma Working Group, Stéphane Adam, Marie Beaulieu, Caroline D Bergeron, Valerian Boudjemadi, Donatienne Desmette, Ana Rosa Danizzetti, Sophie Ethier, Suzanne Garon, Margaret Gillis, Melani Levasseur, Monique Lortie-Lussier, Patrik Marier, Annie Robitaille, Kim Sawchuk, Constance lafontaine y Francine Tougas. 2020 [22]	COVID-19, age discrimination, long-term care homes, seniors.	Oder people	Information gathering	Reduce age discrimination and strengthen the collective in sectors such as health and economy.
Alejandro Klein, 2020 [23]	Gerontology, seniors, COVID-19, thanatopolitics.	Older people in times of pandemic.	Reflection on attitudes and definitions aimed at older people in times of isolation.	Older people do not die only from COVID-19, but from decisions in the absence of deliberate assistance. Importance of the transformation of the health system to avoid discrimination.

**Table 5 ijerph-20-05337-t005:** Articles selected according to area and topic of interest. Source: own authors.

Loneliness and Isolation of the Elderly Throughout the Pandemic				
Authors and Year	Subject	Sample	Methodological Strategy	Results or Main Findings
Ministerio de Sanidad, Servicios Sociales e Igualdad. Instituto de Mayores y Servicios Sociales. 2022 [10]	Multidisciplinary approach to the COVID-19 pandemic.	Workers in the world of care.	Health and gender report 2022.	Strengthen community support to strengthen networks with professionals and the third sector. Change the paradigm of health care based on a hospital-centricvision and enhance primary care actions and services with a community focus. Advance the design and development of a global system of unpaid care.
Cristina M. Pulido, Laura Ruiz-Eugenio, Gisela Redondo-Sama and Beatriz Villarejo-Carballido. 2020 [18]	Social media analysis, network impact, public health, fake news.	Social Media Analytics Reddit, Facebook and Twitter.	Social Media Analysis	False information provided during the pandemic in health networks is carried out aggressively, however the evidence with social impact is respectful and transformative.
Instituto Nacional de Estadística (INE). Notas de prensa. 2020 [24]	Population analysis, ICTs, older people	Survey on equipment and use of information technologies in the homes of people over 65 years of age.	Surveys	Relative tables and population figures, statistical tables on the characteristics of residents.
Ranjan Datta, Jebunnessa Chapola, Prathona Datta and Prokriti Datta. 2020 [25]	Family, resilience, family interaction, COVID-19.	Families during isolation.	Self-ethnography	It highlights the importance of family interaction and resilience in times of confinement.
Johanna Gustavssoon and Linda Beckam. 2020 [13]	Seniors, mental health, COVID-19, health perception	People over 70 years old.	Surveys	Older people comply with the measures imposed. Isolation can lead to worsening mental health and long-term effects.
Manzoor, Shazia. Hamid, Shamikhah. 2021 [26]	Experiences, work-life balance, COVID-19	Experiences of women in the world of work in times of pandemic.	Interviews	Work-related problems, need for tools to telework, family problems related to the care of the elderly and education of children and personal problems such as exhaustion and stress.
Elena Rolandi. Roberta Vocaroo. Simona Abbondanza. Georgia Casanova. Laura Pettinato. Mauro Colombo and Antonio Guaita. 2020 [27]	Communication technology, confinement, loneliness, isolation and social networks.	People 81–85 years old.	Telephone survey	Need to train older people in the use of new technologies to improve their social inclusion.
Fundación Matiaz. 2020 [28]	Seniors, isolation, COVID-19	Older people	Information gathering	Strategies to deal with isolation and protocols to be carried out in the face of the elderly.
Emilia Aiello, Claire Donovan, Elena Duque, Serena Fabrizio, Ramón Flecha, Poul Holm, Silvia Molina-Roldán, Esther Oliver y Emmanuela Reale. 2020 [29]	Social impact and strategies.	Strategies that promote social impact by the social sciences.	In-depth analysis of 6 social science research projects.	Continue to monitor the results obtained to analyse their impact in the future.
Cristina Getson and Goldie Nejat. 2021 [3]	Older people, social assistance robots, health and care, COVID-19, socialisolation, pandemic.	Use of technology in older people during the pandemic.	Interviews	Provide a roadmap for valuing assistance through new technologies.
Seyyed Mohammad Haseein Javadi and Nasim Nateghi. 2020 [30]	COVID-19, older people, psychological effects	Seniors and technological resources.	Reviews	Plan future interventions with elderly groups. Importance of strategic planning.

## Data Availability

Not applicable.

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
