# Peer review of "Ageing and Loneliness in Times of Pandemic: A Scoping Review"

_ijerph, 2023, doi:10.3390/ijerph20075337_

Round 1
Reviewer 1 Report
The article looks solid enough to be published, however, there are some minor considerations I suggest the authors to look into:
-The objective as stated in the abstract, is no clear enough.
-Line 76. This sentence is not clear: "through the collection and / or collection of 76 scientific evidence "
-Line 92. "this group". Has been the group previously defined, is there an age cut?
-In Figure 1 the items go from 10 in step 5 to 22+32 in step 6, which does not add. please review
-I miss in the discussion the comparison with other systematic reviews dealing with the same issue, such as:
Ernst, M., Niederer, D., Werner, A. M., Czaja, S. J., Mikton, C., Ong, A. D., Rosen, T., Brähler, E., & Beutel, M. E. (2022). Loneliness before and during the COVID-19 pandemic: A systematic review with meta-analysis.American Psychologist, 77(5), 660–677. https://doi.org/10.1037/amp0001005
Or, from the methodological point of view:
Ernst, M., Niederer, D., Werner, A. M., Czaja, S. J., Mikton, C., Ong, A. D., Rosen, T., Brähler, E., & Beutel, M. E. (2022). Loneliness before and during the COVID-19 pandemic: A systematic review with meta-analysis.American Psychologist, 77(5), 660–677. https://doi.org/10.1037/amp0001005
Su, Y., Rao, W., Li, M., Caron, G., D’Arcy, C., & Meng, X. (2022). Prevalence of loneliness and social isolation among older adults during the COVID-19 pandemic: A systematic review and meta-analysis. International Psychogeriatrics, 1-13. doi:10.1017/S1041610222000199
-Please review again the whole manuscript, since I found some typos on it.
Author Response
-The objective as stated in the abstract, is not clear enough.
Modified objective. To identify the scientific evidence of the impact that the pandemic has had on the isolation and loneliness of the elderly, from its beginning to the date of search.
-Line 76. This sentence is not clear: "through the collection and / or collection of 76 scientific evidence "
Through the compilation of information, selected after searching different databases, this study aims to establish the impact that the pandemic had on the isolation and loneliness of the elderly. We changed this because of the lack of clarity in the sentence.
-Line 92. "this group". Has been the group previously defined, is there an age cut?
What were the consequences of the pandemic on the experiences, sense of loneliness and isolation produced during confinement amongst the elderly? Because it is a review of articles from different countries, the age range to which we refer has not been specified as each of the countries established different criteria in relation to age groups throughout the pandemic
-In Figure 1 the items go from 10 in step 5 to 22+32 in step 6, which does not add. please review
Figure 1 shows the changes made following suggestions from the reviewer
-I miss in the discussion the comparison with other systematic reviews dealing with the same issue, such as:
The two references are incorporated in the discussion section indicated by the evaluator, lines 32-33. And in the reference section, since we consider their contributions relevant. They are not included in our analysis of results because they did not appear in our initial search in the databases and after incorporating the selection criteria they treated more quantitatively, rather than our objective being more qualitative.

Reviewer 2 Report
This is a study with interesting research questions to address social isolation among older people under COVID-19 pandemic.
I hope my comments below are useful for authors to improve the quality of this manuscript.
1. The objective of the study could be rebuilt to keep consistency throughout the manuscript. Writings in the abstract (L 18), introduction (L75-76), discussion (L254-255) and conclusion (L279-280) are puzzling their research question and objective of the study.
2. Identification of the key words and search equation: Please rationalize how and why the third group (Social and life Or action Or participation or contact or interaction) and the fourth group (active aging) were selected.
Author Response
Reviewer 2
- The objective of the study could be rebuilt to keep consistency throughout the manuscript. Writings in the abstract (L 18), introduction (L75-76), discussion (L254-255) and conclusion (L279-280) are puzzling their research question and objective of the study.
In response to the comments of reviewer 2, also made by reviewer 1, the objective has been redefined in addition to changing sections in the Introduction, discussion and conclusion so that all sections of this work are coherent.
- Identification of the key words and search equation: Please rationalize how and why the third group (Social and life Or action Or participation or contact or interaction) and the fourth group (active aging) were selected.
The keywords were limited to the sociological approach to the aging process that guides this contribution. From the theory suggested by Atchley in 1971 where he asserts that the knowledge acquired throughout the life process of people will be crucial to be able to carry out personal strategies to face, positively, the changes that occur during aging the need to add keywords such as participation, contact, interaction, since these are related to the subject under study.

Round 2
Reviewer 2 Report
Authors revised the manuscript well in line with reviewer’s comments.